# Erosion of Complement Portfolio Sustainability: Uncovering Adverse Repercussions in Steam's Refund Policy

**Samuel Siu** [1,]*[ ], **Yuki Inoue** [2][ ] **and Masaharu Tsujimoto** [3]

[1] School of Accounting and Finance, University of Waterloo, Waterloo, ON N2L 3G1, Canada
[2] Human Augmentation Research Center, National Institute of Advanced Industrial Science and Technology, Chiba 277-0882, Japan, yuki.inoue@aist.go.jp
[3] Department of Innovation, Tokyo Institute of Technology, Tokyo 108-0023, Japan, tsujimoto.m.ac@m.titech.ac.jp
[*] Correspondence: Skcsiu@edu.uwaterloo.ca

**Abstract:** Maintaining a consistently trending portfolio of complements is vital to sustaining platform leadership. Prior research has highlighted the value of open innovation, but has largely disregarded the strategic identification and management of distinctive complements that drive extended platform value, particularly via platform policy modifications. The relevance of prior research around influential policies such as refund leniency becomes largely irrelevant once applied to platform conditions. Utilizing Steam as the medium of analysis, this paper distinguishes complements into three classifications of sustainability, representing its contribution to developing platform leadership. Steam's refund policy alteration is investigated for its effects on refund revenue reductions and additional demand on each classification, assessed using an indirectly related linear regression between playtime distribution and game age, and a binomially distributed *t*-test on the percentage of favorable games. The results reveal that, while all patterns experience significant volumes of refunds, corresponding revenue enhancements are perceived only among unsustainable games. This creates a disadvantageous foundation for high-value complements and consequently, an unforeseen disincentive for association, potentially inciting preferential linkage with competitors. This paper further proposes a precedent for future open innovation and platform management research, where complements of highest relevance are identified and granted heightened priority to protect their sustainability.

**Keywords:** platforms; open innovation; strategic management; platform management; sustainability; complement; complement characteristics

## 1. Introduction

Platform mediated networks have become increasingly common, due to their creation of value in facilitating transactions that would not have otherwise occurred [1]. More specifically, the open business model has enabled platforms to attract valuable products, created by complementors via open innovation, and deliver them to areas of demand [2,3]. Platforms such as Steam have exemplified the value of setting vital platform policies to effectively attract and manage complements and to develop a robust portfolio of complements, thereby enhancing the application of open innovation to help establish platform leadership. Consequently, the focus of this study endeavors to investigate how platform policy design can affect the portfolio of complements in platform-based markets.

The content of the introductory section is as follows. First, the contemporary gaps in platform research are outlined, with reference to the value that this article contributes, mainly in terms of the

identification of the optimal portfolio of complements and the strategic innovations that firms can utilize to manage this portfolio. Subsequently, platform policy design is proposed as a compelling application of innovation management to attract and maintain high-priority complements, improve the facilitation between consumers and complements, and develop platform leadership within the market. The examination is then narrowed to Steam, which is used as the medium of analysis for this study, providing historical and operational background to present vital information for hypotheses and methodology development.

### 1.1. Portfolio Management of Complements within Platform-Based Markets

Prior studies have extensively studied the existence of direct and indirect network effects in various platform contexts [4–7]. This has brought about the belief that the evolution of platform dominance can be explained by the positive feedback loop that occurs within a growing user and complementor base [1]. Other scholars have further extended such studies. While showing evidence of network effects, they emphasize the need to acknowledge other complex factors, implying a tendency for a winner-takes-all model, but by no means a lock-in into platform dominance [8–11]. Research in this area has supported the need for firm-driven strategic innovations to stimulate the development of a platform's dominance and sustainability [12]. For instance, Lee et al. [13] depicted the novel employment of artificial intelligence as an emerging innovative business strategy to improve the quality and function of platforms. Zhu and Iansiti [11] depicted significance in indirect network effects, along with platform quality and consumer expectations, in the video game market. Other studies have confirmed the significance of factors beyond network effects in a diverse range of settings [10,14–17]. This demonstrates the value in successfully managing innovations and the complexity of network markets to redesign a successful platform business model.

Srinivasan and Venkatraman [17] further commented on the insufficiency of taking a black box approach with respect to indirect network effects, where the primary driving factor is excessively fixated on the number of complements. In the framework of the video game industry, they explored various attributes of complements and established significance for platform dominance in a number of titles and genres, the degree of overlap in games, and the extent of complementor status. Other studies have supported the conclusion that, while indirect network effects are present, complements offer differing degrees of value to the platform [18]. This implies that a basic application of open innovation, which purely focuses on the magnitude of each side, may be a suboptimal innovation strategy, prompting the necessity to redirect efforts toward identifying, incentivizing, and managing desirable complements. However, only a small volume of research has addressed this concern, neglecting an imperative possibility of enhancing open innovation research.

While considering the lack of literature around the optimal portfolio of complements, there are surprisingly even fewer insights toward the strategic management and incentivization of complements, outside of a technology management view [1]. Notable exceptions include Venkatraman and Lee's [18] study on the preferential linkages of game creators based on the attributes of game platforms, and Cenamor et al.'s [19] study on the role of in-house complements in accelerating platform adoption. The lack in literature scope has brought about minimal understanding surrounding the tangible strategic innovations that platforms can adopt to foster complement sustainability and supplement open innovation, especially pertaining to the optimal formation of platform policies.

### 1.2. Research Purpose

The application of innovation management within a platform setting can be transposed into the formation of deviceful platform policies to further enhance the potential of open innovation. The purpose of this research is comprised of first seeking to propose a novel approach to the distinction of complements, with a focus on the identification of valuable complements that serve an influential role to establishing platform dominance. While open innovation enables the creation of content via external complements [3,7], this initial step seeks to distinguish complements that can be associated as

centerpiece figures. This is followed by the examination of the refund policy and its corresponding relationship with complements of varying desirabilities, which serves as a highly prevalent and potentially powerful strategic decision that contributes to the cultivation of a robust foundation for vital complements. In doing so, a practical innovation management strategy is examined for its effectiveness in selectively incentivizing the association of the most valuable complements and enhancing the quality and sustainability of the complement portfolio brought about by open innovation. To achieve perception over this, the personal computer (PC) gaming platform, Steam, is employed as the medium of analysis. Two deterrents have hindered the emergence of precedent-serving papers, resulting in little guidance over the optimal approach for analysis. These include the overall lack of literature around strategic management within platform settings [1] and the lack of publicly available data in dominant PC gaming platforms, which has only recently become more widely available from the emergence of player statistic tracking services [20].

Exceptions to studies around the distinction of complements include Bauckhage et al. [20], who sought to cluster games into various playtime patterns, though the paper lacks depiction over the superiority of specific patterns or any meaningful implications concerning innovation management. Despite this, the value of the paper can be conceived from observing differences that exist in game software, beyond simple attributes such as genre or playstyle, which result in various patterns in playtime distribution. This denotes a varying degree of influence that each game has toward the sustainability of the platform, particularly in the ability to maintain its installed base's interest, highlighting the value in proficiently managing a robust portfolio of complements. This paper looks to build on these findings by remodeling the classification approach with the purpose of surveying complement quality in accordance with its ability to support the platform, which is subsequently followed by the analysis of potential tactics to augment complement portfolio quality.

The effective distinction of complements and competent construction of platform policies can contribute to the sustenance of valuable complements. In analyzing the refund policy, this paper seeks to expand the scope of comprehension concerning potential sustainable sources of competitive advantage, particularly as an amplification to the open business model that platforms utilize, which is necessary to ensure a platform's continued dominance over the market [15,21].

### 1.3. The Setting of Steam

Gaming platforms have historically been utilized as a medium for platform research, considering its typically coherent organization of consumers, platform providers, and complements. A substantial portion of platform literature has utilized gaming platforms to empirically demonstrate the existence of network effects [9,11,16,17,19,22]. This article fixates its attention on Steam for reasons including the widespread platform dominance over the PC gaming market, sufficient data availability, and the drastic modifications in platform policies that facilitates the ability to effectively evaluate its aftereffects. To provide some background on the platform, Steam was established in 2003 by Valve corporation, a video game developer, though simply as a method of updating and otherwise supporting the games created by Valve [23]. Steam began transitioning into the contemporary PC gaming platform via open innovation in 2005, with the release of non-Valve games appearing in the Steam game library. Its initial emergence and success can be attributed to "killer" games created by Valve, primarily the release of Half-Life 2 in 2004, that brought about a significant user presence into the platform [23]. With a multitude of users, this established a positive feedback loop of direct and indirect network effects that quickly secured its leadership. Since then, the popularity of previously centerpiece games such as Half-Life 2 has almost entirely declined, though this has been replaced by a large array of other games to keep the platform in dominance, including approximately 30,000 games in its library and 47 million daily active users, as of December 2018 [24].

While the number of games available may provide hints as to the success of the installed base, games that are currently trending are significantly more likely to be the drivers of a sustained installed base. For instance, the top 100 Steam games accounted for more than half of Steam's revenue in 2017

yet comprised less than half of a percentage of Steam's portfolio of games [25]. However, the nature of player behavior results in games rising to popularity and quickly facing downfall, often within a short time span. For example, Steam's top 100 list saw a turnover of over a third of the games from 2016 to 2017 [26–29], illustrating the necessity for a constant stream of new and trending games. Alternatively, possessing a portfolio of lasting games that maintains a strong player base over an extended period can yield a robust degree of platform sustainability. Games such as Counter-Strike Global Offensive and Garry's Mod have held lengthy time spans since their release date, yet continue to maintain renowned positions in the top 100 list since the list's conception in 2016 [26,28,30]. In recognizing this phenomenon, the imperative value in appropriately managing and encouraging the association of these complements that exhibit long-term sustainability is perceived. Strategic mismanagement and the lack of effective innovation tactics can result in a loss of platform dominance, as was seen with past gaming platforms, such as Nintendo Wii, that had negative network effects and positive feedback effects that caused the installed base to quickly dwindle [22].

This paper seeks to build on innovation management and open innovation literature by initiating pioneering contributions regarding major platform policies that have substantial impacts on the success of complements. Specifically, Steam's significant policy change, from a nearly non-existent refund policy to a highly lenient policy [31,32], depicts an unprecedented transition, enabling the examination of behaviors prior to and after the policy modification date and allowing for materialization in comprehension over the effectiveness of the change. This is analyzed in correspondence with games grouped in terms of their degree of sustainability. The analysis of the refund policy is split into three parts. First, games from Steam are categorized according to one of three sustainability patterns, contingent on the timeline of its concurrent user base. While some games may face declining player interest shortly after their release, suggesting an unsustainable pattern, other games see a consistently growing player base over its lifespan. Second, an examination over the policy's effect on each game's existing revenue is conducted. Refund policies are guaranteed to reduce the initial base of revenue, though this depression can either be significant or negligible. Lastly, an analysis over the degree of new revenues is executed, shedding light on whether the refund quantities are sufficiently compensated by the additional income of new players who would not have otherwise contributed to the revenue pool.

The analysis suggests that Steam's redesigned refund policy places sustainable games at a disadvantage in earning revenues, hence, contributing to a decline in complement portfolio quality. More specifically, while existing revenues are reduced to a significant level for all assignments of games, there is little evidence concerning the benefits of additional revenues for sustainable and somewhat sustainable games. This implies a disincentive for these developers to associate with Steam's platform, resulting in a course toward reduced platform quality. The applicability of this study lies in exploring how platform policies can serve as an innovation tactic for reinforcing or hindering the development and success of games, and subsequently drawing implications around understanding the optimal construction of platform policies. The formation of such platform policies sets a robust foundation for the sustainability of platform dominance, notably the incentivization and cultivation of sustainable games, hence, establishing strategies that can be used to manipulate and enhance open innovation. Though a single platform is utilized, due to the prevalence of the refund policy in an abundance of industry and platform settings, general implications can be extended toward a diverse audience. This seeks to contribute to the small but emerging pool of literature around the effective strategic management of complements. Furthermore, this broadens prior research in perceiving the complex dynamics pertaining to the achievement and maintenance of a platform's leadership over its market.

*1.4. Structure of Article*

To provide structure to investigating how policy design can affect a platform's complement portfolio, an overview of the remaining parts of the article is depicted. Section 2 formulates the hypotheses development and methodology of the study. The analysis is framed as a quantitative empirical case study on the effects induced by Steam's alteration of the refund policy, with examination

of multiple facets of the platform to investigate the potential benefits and strains of the refund policy change. More specifically, procedures include the classification of complements into sustainability patterns and the investigation of existing revenue reductions and corresponding revenue enhancements, which are described in Sections 2.5–2.8, respectively. The significance of refunds causing revenue reductions is examined using an indirectly related linear regression analysis between playtime distribution and game age. Meanwhile, the significance of additional revenue flow is evaluated using an application of central limit theorem to construct a binomially distributed *t*-test involving the comparison of the proportion of games that exceed extrapolated expectation with a random null hypothesis.

Section 3 shares the results and hypotheses verification of the methodology around existing revenue reductions and additional revenue flows for each sustainability pattern, illustrating potential benefits to unsustainable games, but adverse consequences to high-value sustainable complements. Section 4 concludes the results by extrapolating important discussion and application to the study, exemplifying the potential repercussions of poor policy design and conveying how this can hinder the future sustainability and leadership of platforms such as Steam. In doing so, valuable insights are obtained around how platform providers should pursue the design of the refund policy, or more generally, the development of platform policies, so as to ensure the platform's continued dominance.

## 2. Materials and Methods

### 2.1. Refund Policy Theory

Prior research has analyzed the interplay among quality, pricing, and the return policy [33]. For instance, Li et al. [34] argued that under conditions of high-quality products, optimization of profits can be achieved through expanding leniency by virtue of lower volumes of refunds and quality signaling. Though such research may provide insight in a retail setting, only partial relevance can be taken in the analysis of network markets, particularly in the setting of Steam. For instance, aspects such as signaling are largely unreliable for consumers, in view of the lack of significant control that Steam exercises over the quality of complements. This is further amplified by Steam's updated game admission policy, simply requiring payment as the pre-requisite for publishing [35], and their recent post claiming minimal responsibility over the content inside the games published [36]. Likewise, while the quality of a retail product is often correlated to its degree of usefulness and durability in accomplishing its respective function, game quality only partially determines player satisfaction [37]. Gonzalez et al. [38] discussed the vast differences concerning typical product usability and game playability before outlining a comprehensive definition of playability, incorporating elements such as the quality of gameplay and storyline and the degree of control and realism, among other aspects. In short, while high quality products are very likely to fulfill their promoted function, and thus, face a low number of refunds, high quality games may still carry deficiencies in fulfilling consumer tastes [34]. Despite the differences, some of the prior research remains relevant.

For example, studies have demonstrated that refund policies theoretically generate more demand, constituting the primary rationale for the policy [33,34]. The value of the policy has predominantly served as a primary motivator for a consumer's purchase decision [33], on the grounds of the flexibility of reversing a regrettable decision [39]. In a survey of U.S. online customers, 22% of customers claimed that the non-existence of a refund policy would sufficiently hinder their purchases from a respective vendor. Likewise, 72% believed that a policy that did not require the communication of a reason would increase the likelihood of a purchase [40]. Wood [39] further supported these conclusions, depicting significance over a lower deliberation time under refund policy conditions. While sale increases may be evident, an important element in the optimization of the leniency of a refund policy originates from the fact that a more lenient policy increases the probability of return, and hence, raises costs and reduces profits [33,41]. Davis et al. [41] discussed the optimization of hassle-like conditions, which reduces this probability and can potentially prevent customer abuse over the policy. The application of this

research is applied to examining Steam's refund policy, particularly its effectiveness in supporting groups of games, classified into their respective degrees of sustainability.

*2.2. Hypotheses Building*

Steam's unparalleled shift in the leniency of their refund policy allows for the comparison of settings among various stakeholders. Formerly, Steam disallowed refunds upon completion of the download of games, with few exceptions provided unless under exceptional scenarios [32]. The transition of policies on June 2, 2015 provided significant leniency to consumers, granting eligibility for any reason under the conditions that the game was purchased within the preceding 14 days and played below 2 h [32]. Steam further elaborated upon the interpretation of the all-inclusive "any reasons", providing examples including repurchasing at a discounted price and simply disliking the game [42].

The leniency of Steam's refund policy, particularly its 2-hour playtime threshold and effortless refund procedures [43] sets up an excessively high probability of refund. This is further extended by the difficulty in pleasing the interest of all gamers, suggesting an elevated likelihood of refunds from a segment of players. Even among gamers whose tastes are appeased, the 2-hour threshold is beyond what many would view as an appropriate trial time, highlighting the reality that a player could complete a significant portion of the game and still be granted eligibility for a refund. In a survey of 220 Steam game developers, only 34% were in support for the highly uniform 2-hour threshold across all games [44]. From a higher-level view on Steam's policy management, there is little variation in the favor of complementors, with a mere 24% believing that Steam sufficiently tended to their questions and concerns, and only 22% believing that Steam carried interests aligned with theirs [45]. This underlines the lack of communication and resolution over issues raised, suggesting that innovations via new policies could potentially damage complements. Based on such findings, the following hypothesis is proposed.

**Hypothesis 1.** *The existing revenue pool for games in all classifications will be significantly and adversely affected by the transition to the new refund policy.*

The intention of permitting leniency in refunds is to generate increased demand. Players may find themselves in greater willingness to experiment with games and diminished pressure to deliberate over purchases, considering that the uncertainty is mitigated by the possibility of a reversal [39]. However, this lacks alignment with statistics pertaining to the lack of favor among complementors. In reconciling this, it is possible that while complementors undergo a net benefit, their discontentment stems from the belief that the policy can be further enhanced. Since Steam receives a 20–30% split in revenues with complementors [46], maximizing revenues would favor both parties. Based on this argument, the following hypothesis is put forward.

**Hypothesis 2.** *The additional revenue pool for games in all classifications will be significantly and positively affected by the transition to the new refund policy.*

The remainder of this paper endeavors to confirm these hypotheses, supplying perception to the effectiveness of Steam's innovative modification to the platform policy.

*2.3. Methodology*

The following sub-sections outline the methodology, which is separated into three steps. The first step seeks to recognize distinctive attributes that complements portray by enforcing classification based on complement quality. Accordingly, games are classified into three respective groups: sustainable, somewhat sustainable, and unsustainable. While this does not address a specific hypothesis, the following two steps are applied to each group separately, yielding comprehension over the

difference in outcomes induced by the policy alteration among the three groups. The second step examines the significance of reductions in existing revenue, seeking to understand whether refund quantities are considered significant or negligent and addressing matters described in Hypothesis 1. The third step examines the significance of additional demand generated, looking to verify Hypothesis 2. Through consolidating the effects observed in Hypothesis 1 and Hypothesis 2, greater understanding can be achieved over the favorability and net benefits of the policy change. Figure 1 provides a brief overview of the procedures conducted, as well as the data utilized to perform each analysis. The rest of the methodology section explains the procedures in greater detail, along with the rationale behind the selection of procedures employed.

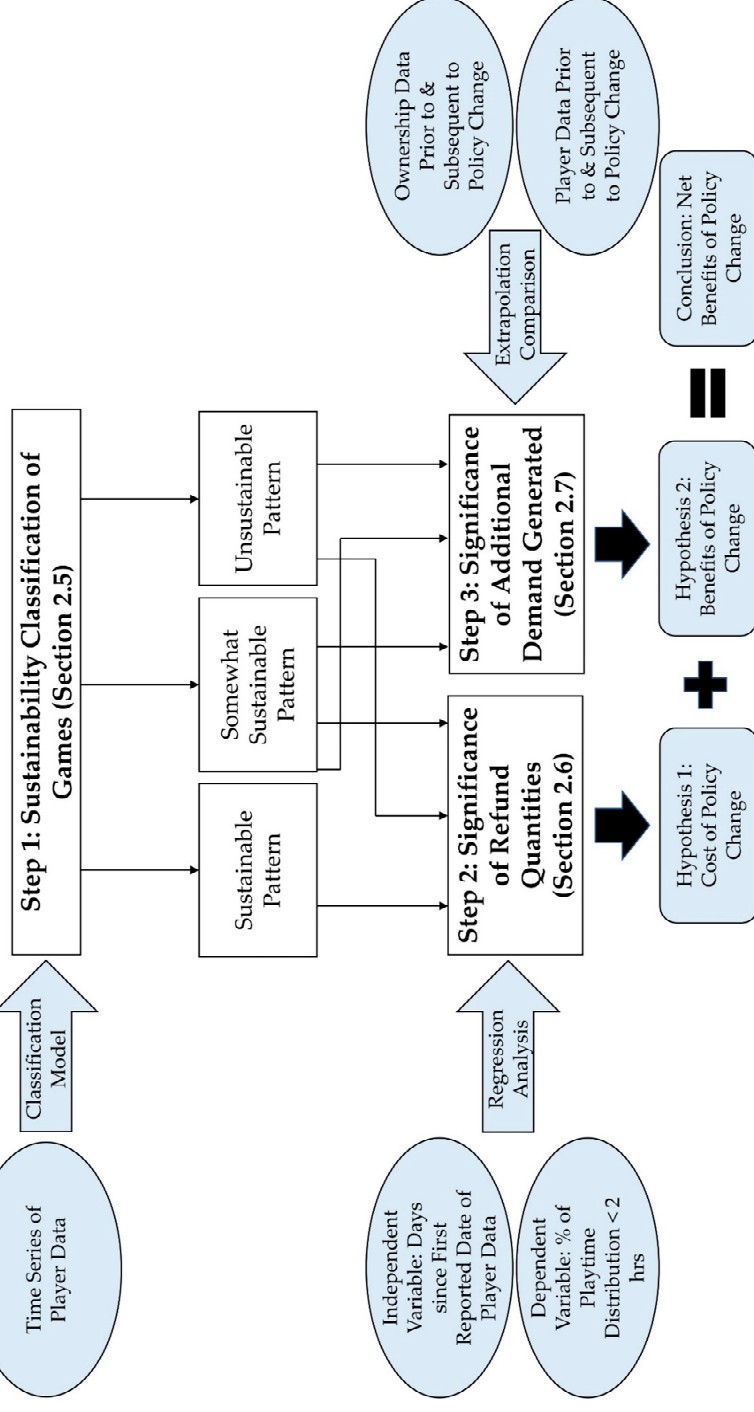

**Figure 1.** Conceptual overview of the model and related data application.

### 2.4. Data

To perform this study, as consistent with Bauckhage et al.'s [20] analysis, data is collected via player statistic tracking services, specifically Steam Database and SteamSpy. Steam Database accumulates data directly from Steam's website and update system [47]. SteamSpy formerly derived their data by means of analyzing a sample of publicly available game libraries of players. Succeeding Steam's privacy policy change in April 2018, which hindered this approach, an alternative method designed around machine learning was utilized [48].

The sample consists of Steam games with quantity of owners above 750,000 as of 24 November 2018. This yields a sample size of 678 games, of which the series of daily data for concurrent players and ownership quantities since release date is retrieved for each game. Data for corresponding analysis is accessed up until, and including, November or December 2018. Additional information regarding data collection is provided in this article's Supplementary Materials. Several properties and limitations further diminish the sample size available for examination. First, several games released during and prior to 2008 lack data sufficiency, particularly near launch date. This sample is further confined to all games with a price tag, as would be most appropriate in examining the refund policy. A sample size of 422 games remains, which is adopted during the first two steps of methodology. Since the third step highlights a comparison between the former and latter state of the refund policy alteration, only games that possess data within a minimum of one week prior to 2 June 2015 are studied, leading to a sample size of 326 games. See Appendix A for discussion in respect of the appropriateness of the sample size employed.

### 2.5. Classification of Sustainability Patterns

Games are first classified according to one of three groups: sustainable, somewhat sustainable, and unsustainable. To achieve this, data for each game is graphed along a time-series graph. The independent variable ($x$) represents the number of days since first day of reported data. The dependent variable ($P$) represents a quantity that demonstrates the popularity of the game. While ownership data provides little information concerning the relevance of games at any selected point of time, daily concurrent users serve as a superior indicator in illustrating the prevalence of a game. Therefore, a preliminary classification model revolving around daily concurrent users, or player base ($P$), is designed.

The data proved to possess substantial difficulties, due to the presence of zeros, implying a lack of data, as well as days of irregularly low data, perhaps suggesting downtime or other special circumstances. Further sources of volatility within concurrent user data include the presence of free weekends, allowing for the trial of a game for typically 3 days, as well as periodic game discounts and in-game events. To ensure that only the most relevant and reliable data is applied to the model, efforts are taken to mitigate the effects of exceptional data. To perform this, the 30-day maximum average is first calculated based on raw data, represented by the variable, $\delta$. Data that is below 10% of the largest 30-day moving average ($0.1\delta$) is removed. Subsequently, a polynomial best-fit trendline is placed, with coefficients a, b, and c.

$$P = ax^2 + bx + c \tag{1}$$

From the first point of data where the threshold of $0.1\delta$ is met, to the last point of collected data, the average of the first half of dates is compared to the latter half. Utilizing the coefficients of the polynomial trendline and a comparison between the former and latter half of the data, the games are preliminarily classified into the three groups of sustainability. Table 1 describes the four different possibilities and its related classification, while Figure 2 depicts an ideal illustration of the trendline constructed for each scenario.

**Table 1.** Description of Classification Possibilities and Related Sustainability Pattern Association.

| Figure No. | Description | Classification Pattern |
|---|---|---|
| a | a < 0, mean of first half < mean of second half | Sustainable |
| b | a < 0, mean of first half > mean of second half | Somewhat Sustainable |
| c | a > 0, mean of first half < mean of second half | Sustainable |
| d | a > 0, mean of first half > mean of second half | Unsustainable |

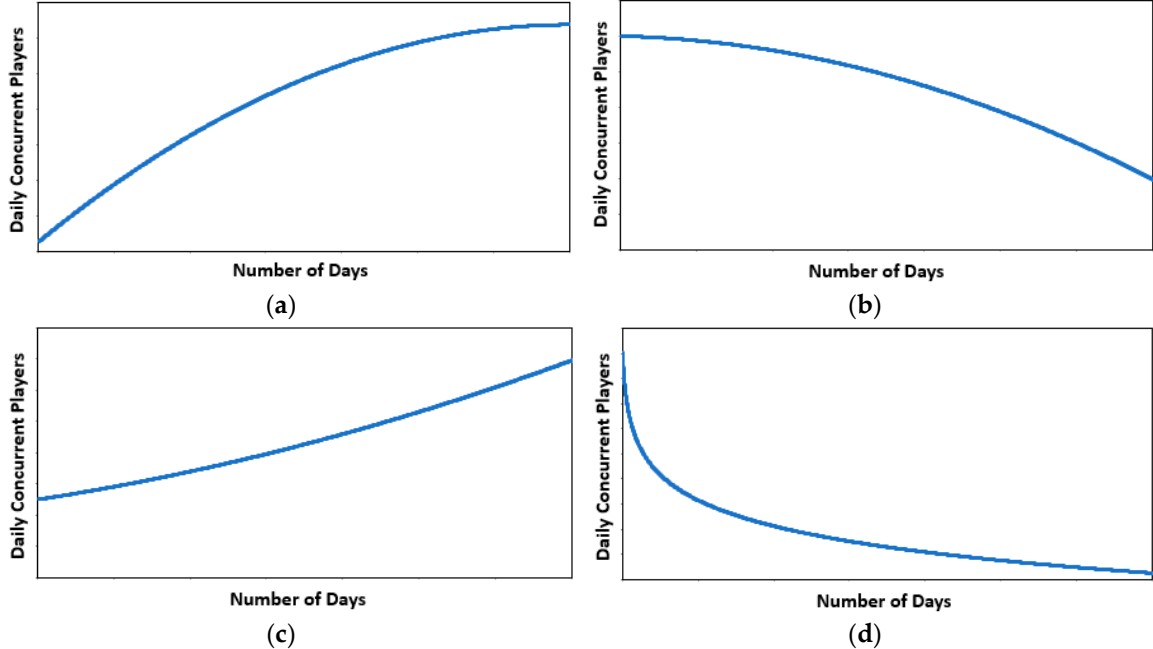

**Figure 2.** Illustration of ideal time-series trendline results under each classification possibility: (**a**) Sustainable (**b**) Somewhat Sustainable (**c**) Sustainable (**d**) Unsustainable.

Under a sustainable game pattern, a game is capable of sustaining and increasing its player base over an extended duration. Growth is maintained for a substantial period before potentially reaching a standstill and a relatively slow decline. Conversely, unsustainable games attain high popularity at launch but experience immediate and alarmingly fast decline, demonstrating a lack of ability to maintain its initial player base. Somewhat sustainable games fall between these two extremes. For instance, while there may be decline almost immediately after release, the decline tends to be significantly slower. Another possibility occurs where the player base grows after release, though only for a short time, before falling in popularity.

However, under this classification model described, several scenarios can lead to various unintended classifications. Appendix B provides details regarding additional procedures to address these issues.

While this approach somewhat disregards the absolute length of a game's existence and the magnitude of the player base, utilizing a relative comparison of games allows for the assessment of sustainability in each game's tier of success. In other words, a highly popular game with a large quantity of owners does not necessarily represent a sustainable game, especially if it demonstrates an unsustainable player–time graph. Conversely, a significantly less popular game may still be assessed as sustainable if it can successfully preserve its player base. Hence, an absolute measure possesses the limitation of becoming significantly biased toward popular games. To allow for the comparison and classification among games that differ vastly in degrees and lengths of popularity, a relative comparison is utilized.

Despite this, the outlined model could potentially become biased toward recently launched games. For instance, following an excessively large duration of time, any game would eventually

decline to a stage of obsolescence. At this point, the average of data values existing in the former half would more likely exceed the latter half, leading to a bias toward a lower sustainability classification. Appendix C addresses this concern by portraying the distribution of game age for each sustainability group, showing the unlikelihood of significant discrepancy by this limitation.

Subsequent steps apply identical procedures to each sustainability pattern, providing insights pertaining to the two sides of resulting consequences from the policy innovation.

### 2.6. Significance of Reductions in Existing Revenue

Ideally, the optimal analysis to assess the reduction of revenues incorporates the quantity of refunds as an absolute number or relative percentage of sales. The lack of publicly available data, which can be partially attributed to Steam's agreement with developers and prohibits the distribution of detailed sales data [44], necessitates an alternative channel. Substitute approaches can be composed by examining data that is substantially affected by the acceleration of refund quantities. This paper exploits the indirect effects on playtime distribution to pinpoint significance of refunds.

Steam's policy change enabled players to successfully request a refund under the conditions of a playtime below 2 h and a purchase within the previous 14 days. Therefore, a rational player would theoretically seek a refund for a game if conditions are fulfilled and intentions to continue playing are inadequate. If this group is substantial, in light of the eligibility for refunds that owners now possess but previously lacked, complements would likely perceive a reduced proportion of owners with playtime distribution below 2 h. Considering this, the post-policy change period, or the period after the policy alteration, would encompass a reduced proportion of owners with playtime distributions below 2 h, assuming the effects are significant. The period prior to the policy change should occupy a higher proportion of playtime distribution below 2 h.

Based on the differences that should be observed between the period prior to and following the policy modification, a game that is launched at an earlier time should see higher proportions of playtime distribution below 2 h, based on the logic that it has resided in the pre-refund policy change period for a more prolonged period, all else being equal. Table 2 further illustrates this concept, where the pre-refund policy period holds a proportion of 10%, compared to the post-policy change proportion of 5%. Two observations are highlighted from Table 2. First, an older game tends to carry a higher proportion of owners with playtime distribution under 2 h. This is consistent with the fact that these games occupy a higher fraction of lifespan in the pre-policy change period. Second, any game released after the refund policy alteration should possess the same proportion, denoting a lack of correlation among these games. Such games carry lifespans entirely situated in the post-policy change period, and hence, have always had a reduced proportion of playtime distribution below 2 h. This is vital to eliminating the possibility that time could be the underlying factor around the reduced proportion. Under this scenario, the declining percentage may merely be caused by a growing trend for players to fix their time on specific games, inducing shifted playtime distributions as years progress.

**Table 2.** Hypothetical Illustration of Relationship between Game Release Date and Playtime Distribution below 2 h, Assuming Significance of Refund Quantities.

| Game Release Date (First Day of Data) | 2012 | 2013 | 2014 | 2015 | 2016 | 2017 | 2018 | Average |
|---|---|---|---|---|---|---|---|---|
| 2012 | 10% | 10% | 10% | 5% | 5% | 5% | 5% | 7.14% |
| 2013 | | 10% | 10% | 5% | 5% | 5% | 5% | 6.67% |
| 2014 | | | 10% | 5% | 5% | 5% | 5% | 6.00% |
| 2015 | | | | 5% | 5% | 5% | 5% | 5.00% |
| 2016 | | | | | 5% | 5% | 5% | 5.00% |

In line with Table 2, a linear regression is constructed between the independent variable, number of days between the release date and the policy change date ($x$), and the dependent variable, proportion of owners with playtime distribution below 2 h ($P$). The independent variable is a measure of age,

which is positive for all games released before the policy transition and negative for games released afterwards. As previously explained, if the age of the game is positively related to the proportion of playtime distribution below 2 h, the significance of refund quantities is considered verified.

However, as the presence of zeros within player data highlights the existence of missing data, to ensure consistency with the formation of playtime distribution by the data source, SteamSpy, the number of days since the first day of reported player data, instead of the release date, is utilized as the independent variable. Equation (2) portrays this relationship, where the coefficient, s, comprises the element of interest, representing the effects that age, or more accurately the number of days since first reported data (x), elicits on the proportion of playtime distribution below 2 h (*P*).

$$P = sx + D \tag{2}$$

### 2.7. Significance of Additional Revenue Flow

To investigate the significance of additional revenue flows, a comparison between expected and actual values is executed on player base and ownership data. Expectations are generated by employing a linear trendline on the 30 days prior to the policy change date. This is subsequently applied to extrapolate the following three 30-day periods, for a total extrapolation of 90 days. Figure 3 illustrates this analysis. The red portion represents the formation of expectations using the 30 days prior to 2 June 2015, while the black portion represents a 90-day extrapolated period. Given the shape of the data, expectations conceived through a polynomial trendline may appear more appropriate. However, under conditions of higher volatility, which is prevalent among the data for many games, quickly descending or ascending polynomial trendlines may be formed, leading to unrealistic and irrelevant expectations.

Expectations devised based on a 30-day period balances the limitations from employing an excessively protracted or abbreviated period. An overly lengthened period lacks reliability in extrapolating for the near future, as the date immediately before the policy change date may not necessarily correspond to the lengthy trendline, considering the high volatility of the data. Conversely, an excessively short period is vulnerable to daily fluctuations, particularly from weekends within a weekly cycle.

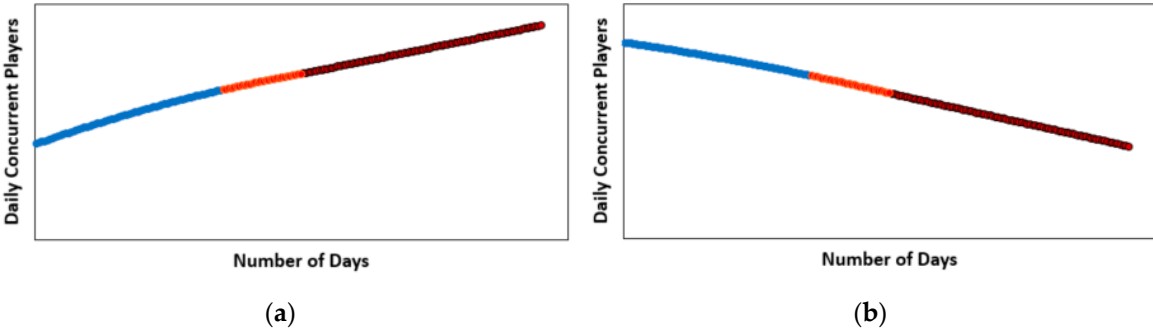

(**a**)    (**b**)

**Figure 3.** Illustration of extrapolation of 30-day period preceding policy change to obtain expected values for 90 days following policy change: (**a**) Expectation formation—incline (**b**) Expectation formation—decline.

Moreover, the presence of free weekends could drastically disturb the formation of expectations or the comparison of the prevailing player base. Such events cause significant hikes in the short-term before generally reverting to its previous state. Figure 4 depicts examples of this effect, paralleling Figure 3 but incorporating a free weekend immediately preceding the policy change date. The result is the formation of vastly different and substantially irrelevant expectations.

While the methodology employed to collect concurrent daily players is publicly available via Steam, thus, is reliable under stress conditions such as free weekends, ownership data is estimated

by analyzing a brief duration of historical playtime. Hence, although the accuracy of player data is maintained during free weekends, ownership data is substantially misrepresented for a short period after the event. Considering this, the elimination of data is varied for player data and ownership data. Under player data, data is eliminated between one day preceding and 30 days succeeding the free weekend. Conversely, with ownership data, data is eliminated when an unusually high increase is first observed up until 75 days after the free weekend. The unusually high increase, which commences the elimination of data, is highly recognizable and encompasses little ambiguity in identification.

However, an important consideration involves the necessity to examine the model's reliability when extrapolating over extended durations. In other words, the elimination of data entails an extension of extrapolation to accommodate affected periods of data, yet extrapolation via a linear trendline is only most effective under short periods. To balance the reliability of projected values, the span of analysis is limited to 60 days prior to (3 April 2015) and 150 days after (29 October 2015) the policy transition. Where this conflicts with the elimination of data and prohibits the examination of a full 30-day period, the duration is shortened accordingly to a minimum length of 7 days. This seeks to achieve a balance between removing the effects of free weekend fluctuations and ensuring the reliability of extrapolated values.

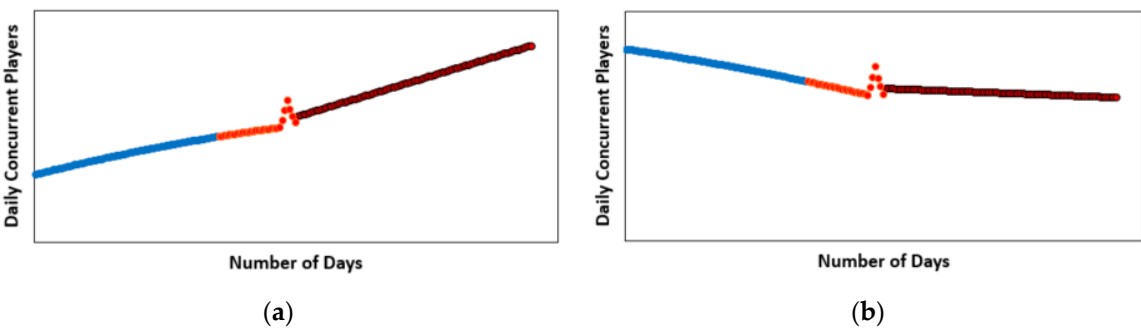

(**a**)　　　　　　　　　　　　　　　　　　(**b**)

**Figure 4.** Illustration of extrapolation of 30-day period preceding policy change to obtain expected values for 90 days following policy change, with presence of free weekend immediately prior to policy change: (**a**) Effects of free weekend—incline (**b**) effects of free weekend—decline.

## 2.8. Measure of Significance on Additional Revenue Flow Procedures

Due to the lack of a regression employed, significance levels must be measured with alternative methods. In comparing whether one set of data is significantly enhanced relative to another set of data or null hypothesis, approaches such as *t*-test and ANCOVA can be employed [49] (p. 354). However, the relevance of such tests when conducted on multiple games is relatively diminished. For instance, in a hypothetical example of 200 games, if 150 games, or 75% of games, are to display significant enhancements, questions remain unanswered around the general impact of the policy transition on the group of games. In other words, such methods lack a threshold to establish significance when working with multiple sets of time-series data. On the other hand, by utilizing a binomial distribution of outcomes, an application of the central limit theorem, which enables the distribution of percentages of favorable outcomes to be depicted as a normal distribution, it potentially enables the comparison to a random null hypothesis using a binomially distributed *t*-test, providing basis for the establishment of significance as to whether the policy has benefited the group, in view of the entire sample size employed [49] (pp. 243–244). In respect to the binomial distribution, only two outcomes are possible, consisting of "favorable", where actual values exceed extrapolated expectations, and "unfavorable", where actual values are inferior. Significance can be measured based on whether the binomial distribution outcome is random or significantly skewed toward the favorable outcome.

Still, concerns may be raised in relation to whether the magnitude of the favorability is truly significant at an individual game level. Appendix D further discusses this matter, with application of

the *t*-test method, showing that an abundant majority of games that fulfill the "favorable" outcome are also significant from an individual analysis standpoint.

Thus, significance is measured through two independent approaches, both of which apply the central limit theorem to construct normal distributions. Considering that any other indiscriminate date should possess a 50% likelihood of favorable results, implying randomness in the outcome, a null hypothesis for disproval purposes is established. The first approach calculates the likelihood of a value equal to or of higher favorability than the computed results (See Tables 5 and 6 in Section 3.3 for computed results), assuming the probability of a favorable result is random at 50% probability. The second approach estimates a percentage equal to the computed results (See Tables 5 and 6 in Section 3.3 for computed results) as the probability of a favorable result. The likelihood of a hypothetical result equal to or of lower favorability than 50% is subsequently calculated based on the distribution created. Both approaches utilize a one-tail distribution for determining significance levels. The computation of the standard deviation ($\sigma$) is depicted below according to the central limit theorem, where $n$ represents the sample size and $P$ represents the probability of assumed success [49] (pp. 240–243).

$$\sigma = \sqrt{\frac{P \times (1-P)}{n}} \tag{3}$$

## 3. Results

As previously discussed, effects from the policy transition evokes two sides of effects, which should be considered in correspondence to one another. This section first investigates the significance of existing revenue, before proceeding to the analysis of additional revenue flows. To analyze changes in existing revenues, a high-level examination, incorporating all games, is first conducted before the regression on each pattern is isolated.

### 3.1. High-Level Analysis of Reductions in Existing Revenue

Table 3 provides a summary of high-level regression results of the relationship between age, the independent variable, and proportion of playtime distribution below 2 h, the dependent variable, where sustainability groupings are disregarded. Model I conducts a comprehensive regression among all games, showing significance at the 1% level, demonstrating that preceding games possess a higher proportion of owners with playtime distribution below 2 h. Figure 5 illustrates the linear regression graph from Model I. Two data points are conspicuously separated from the cluster of data, lacking coincidence with the composed trendline and advocating for its elimination due to outlier features. Both games possess distinctively high ages, being released over 3500 days prior to 2 June 2015, and they are of Steam's originating games. Under these circumstances, Steam users would likely have focused on these games, considering that the games were highly popular when the platform's game library was scarce, ultimately producing a shifted playtime distribution. Furthermore, as discussed in Appendix C, highly mature games are significantly more likely to be inclined toward a lower sustainability pattern. For these reasons, both data points are considered outliers and eliminated for further analysis.

Model II confines the data to games released after the policy change. As aligned with expectations, the two variables exhibit no significant relationship ($p = 0.87$) and a descending trendline with slope of $-0.0007635$. This eliminates the possibility that time is the underlying cause for the significant relationship observed in Model I. Model III applies filtering to games with a median playtime below 3 h. While a positively sloped trendline is examined, the slope is notably lower at 0.001338, with no significance observed ($p = 0.396$).

**Table 3.** Non-Pattern Specific Linear Regression Analysis Results Illustrating Significance of Refund Quantities Induced by Refund Policy Alteration.

| Model | I | II | III |
|---|---|---|---|
| Intercept | 25.3179 *** | 21.0946 *** | 50.9961 *** |
| | (0.9376) | (2.6276) | (2.0721) |
| Slope | 0.00635 *** | −0.0007635 | 0.001338 |
| | (0.000833) | (0.004664) | (0.00157) |
| Number of Observations (N) | 422 | 95 | 97 |
| *R* squared | 0.1215 | 0.0003 | 0.0076 |

Note: Independent variable—game age, Dependent variable—proportion of playtime distribution below 2 h. Model I—comprehensive regression, Model II—filtered by games released after policy change, Model III—filtered by games with median playtime below 3 h. *** $p < 0.01$.

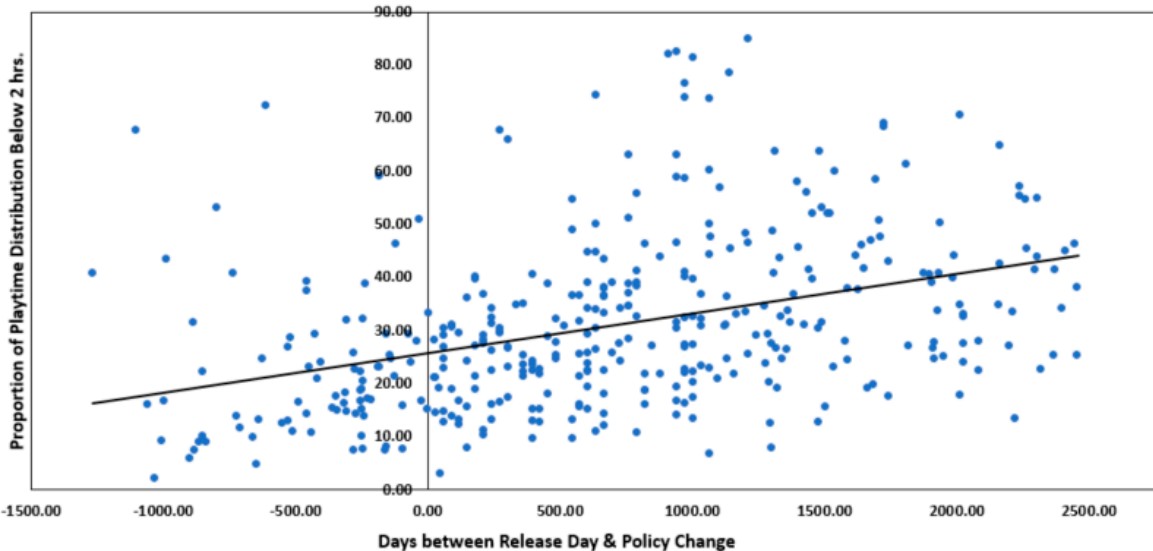

**Figure 5.** Comprehensive linear regression analysis between game age and playtime distribution below 2 h (Model I).

### 3.2. Detailed-Level Analysis of Reductions in Existing Revenue

Table 4 summarizes results for Model IV to VIII, describing the linear regression analysis between age and playtime distribution below 2 h for each pattern of sustainability. Model IV depicts the results for the sustainable pattern, showing a positive slope of 0.003755 with a significance level of 10%. The slope exhibits the positive effect that age has on the proportion of playtime distribution below 2 h. This can be attributed to significant refund quantities, causing a diminished proportion of owners with playtime distribution below the 2-hour threshold following the policy modification. Model V illustrates the linear regression among somewhat sustainable games. Though a positive slope of 0.003476 is perceived, significance is not achieved. However, as previously observed in Model III, the volatility of games with low median playtime can contribute to the concealment of significant relationships. Hence, Model VI refines this analysis by confining the sample to games with median playtimes above 3 h. Consequently, the slope is considerably increased to 0.0046, with significance level of 5% attained. Model VII portrays the relationship among unsustainable games, conveying a slope of 0.00748 and a significance level of 1%. As consistent with the procedures of Model VI, Model VIII similarly removes games that do not satisfy the 3-hour median playtime threshold. While the same level of significance is reached, a substantially lower slope of 0.00503 is recognized, though this is complemented by an increase in the coefficient of determination. No games within the sustainable game pattern display a playtime median below 3 h, which would appear reasonable considering their capability to sustain player interest.



**Table 4.** Linear Regression Analysis Results on Individual Sustainability Patterns, Illustrating Significance of Refund Quantities Induced by Refund Policy Alteration.

| Model | IV | V | VI | VII | VIII |
|---|---|---|---|---|---|
| Intercept | 12.0988 *** | 24.907 *** | 16.0762 *** | 25.7163 *** | 20.7652 *** |
| | (1.8186) | (3.8048) | (2.4291) | (0.9956) | (2.6028) |
| Slope | 0.003755 * | 0.003476 | 0.0046 ** | 0.00748 *** | 0.00503 *** |
| | (0.00189) | (0.002664) | (0.001694) | (0.000921) | (0.000605) |
| Number of Observations (N) | 24 | 23 | 16 | 375 | 278 |
| R squared | 0.1583 | 0.0784 | 0.345 | 0.1504 | 0.2005 |

Note: Independent variable—game age, Dependent variable—proportion of playtime distribution below 2 h. Model IV—sustainable games, Model V—somewhat sustainable games, Model VI—somewhat sustainable games with median playtime above 3 h, Model VII—unsustainable games, Model VIII—unsustainable games with median playtime above 3 h. * $p < 0.1$, ** $p < 0.05$, *** $p < 0.01$.

### 3.3. Analysis of Additional Revenue Flows

The reduction of existing revenues places complementors at a disadvantage only if adequate additional revenues are not achieved. Table 5 summarizes results concerning the proportion of games that exceed extrapolated expectations, utilizing ownership data. Based on summary results, all sustainability patterns display a significant percentage surpassing 80% under the 30-day period after the policy alteration. The following two 30-day periods show steadily declining percentages, though all percentages remain above 60%. Among sustainable and unsustainable games, all results are significant to the 1% level. In contrast, for somewhat sustainable games, the significance level quickly drops from the 1% level to an insignificant level throughout the progression of the three 30-day periods.

Table 6 depicts relatively contradictory results among sustainable and somewhat sustainable games. While the unsustainable pattern conveys benefits to the 1% significance level under all periods analyzed, only sustainable games share in the benefits to a relatively low significance level of 10% during the first 30-day period. Somewhat sustainable games possess a percentage exceeding the 50% threshold during the first 30-day period, though no conclusive confidence interval is achieved. Following the 30-day periods, both patterns hover around 30 to 40%, which is subordinate to the random 50% null hypothesis threshold. As the player base cannot logically be adversely affected by this policy change, this should be interpreted as a lack of additional players instead.

**Table 5.** Summary Results of Percentage of Games Displaying Ownership Data Increases Beyond Extrapolated Expectations after Refund Policy Alteration.

| Pattern of Sustainability | Sustainable | Somewhat Sustainable | Unsustainable |
|---|---|---|---|
| 30 days—% of Games Above Exp. | 94.44% *** | 84.62% *** | 82.94% *** |
| 60 days—% of Games Above Exp. | 88.89% *** | 76.92% ** | 76.19% *** |
| 90 days—% of Games Above Exp. | 77.78% *** | 61.54% | 71.77% *** |
| Number of Observations (N) | 18 | 13 | 294 |

Note: Significance of additional revenue flows assessed through binomially distributed *t*-test. ** $p < 0.05$, *** $p < 0.01$. Both approaches of measuring significance yield the same significance level.

**Table 6.** Summary Results of Percentage of Games Displaying Player Data Increases Beyond Extrapolated Expectations after Refund Policy Alteration.

| Pattern of Sustainability | Sustainable | Somewhat Sustainable | Unsustainable |
|---|---|---|---|
| 30 days—% of Games Above Exp. | 66.67% * | 61.54% | 81.02% *** |
| 60 days—% of Games Above Exp. | 38.89% | 33.33% | 60.96% *** |
| 90 days—% of Games Above Exp. | 33.33% | 33.33% | 57.53% *** |
| Number of Observations (N) | 18 | 13 | 295 |

Note: Significance of additional revenue flows assessed through binomially distributed *t*-test. * $p < 0.1$, *** $p < 0.01$. Both approaches of measuring significance yield the same significance level.

*3.4. Explanation of Results and Verifications of Hypotheses*

In analyzing the impact on revenue reductions, significance is observed in all three patterns of sustainability. This demonstrates the prevailing curtailment of playtime distribution under 2 h during the post-policy change period. As discussed, this is indirectly caused by the presence of refund quantities due to the 2-hour playtime threshold for refund eligibility. This consequently causes a proportion of owners within the 2-hour playtime threshold to be removed from the playtime distribution construction.

The significance level in sustainable and somewhat sustainable games may be considered substantially low, at 10% and 5%, respectively. This can be partially traced to the low sample size available, which is rationalized in Appendix A. Conversely, the robust significance level recognized in unsustainable games can be associated with the noticeably high sample size relative to other patterns. As described in Model II of Table 3, further tests were conducted to disprove time as the underlying factor at a comprehensive level of regression analysis. The negative slope and lack of significance pinpoints the refund policy change as the prevailing cause for the relationship. Inadequate sample size, particularly among the sustainable and somewhat sustainable pattern, hindered the performance of this test on individual classification groups. With respect to this, Hypothesis 1 is considered verified, as all three patterns of sustainability show a significant relationship between game age and playtime distribution below 2 h.

The analysis pertaining to the influence on additional generated revenues employs two separate sets of data, yet brings seemingly contradictory findings, particularly among sustainable and somewhat sustainable games. In reconciling the supposedly conflicting results prevalent in Tables 5 and 6, attributes of the two sets of data should be recognized. While data points under player data are computed independently from each day, ownership data involves a consolidation of all prior data points. Under the results for player data, favorable results that exceed the 50% random threshold are perceived during the first 30-day period, stimulating an incline in ownership as well. This can be attributed to an initial period of "hype", induced by the introductory excitement from the announcement and the vastly advantageous modification to refund leniency. Little evidence of sustained advantages is observed after the initial 30-day period among sustainable and somewhat sustainable games. Despite this, the pioneering enhancement in ownership could generate a delay in results, causing inconsistencies with player data. This is further supported by the depreciating progression of percentages as each ensuing 30-day period in ownership data is observed, portrayed in Table 5. For instance, the decline in percentage between the first (30-day) and third (90-day) 30-day period is relatively high for sustainable and somewhat sustainable games, at 16% and 23% respectively, compared to the 11% recession seen in the unsustainable pattern. This decline can be explained by two reasons. First, as the period of extrapolation is extended toward an increasingly lengthy period, expectations drift in reliability, resulting in higher randomness and a tendency to gravitate toward a random 50% percentage. Manifested only in sustainable and somewhat sustainable games though, the combination of a lack of additional flow of owners and a significant magnitude of refunds evokes a decline in games exhibiting favorable outcomes. This resultantly brings upon a more rapid decline in percentages. Based on these results, Hypothesis 2 is only considered proven for unsustainable games.

## 4. Discussion

*4.1. Explanation of Results and Verifications of Hypotheses*

Prior studies have extensively evaluated various contributing factors to platform dominance, including the necessity for a robust user and complement base [14–16], the benefits of platform quality [10,11], and the value of specific characteristics in technology architecture [1,50]. A small proportion of emerging research has examined the preferential linkage that complementors demonstrate [18], as well as the optimal portfolio of complements [17,19,22], sharing potential areas of improvement within the scope of open innovation.

However, these areas of research offer inadequate and broad applications to platform-mediated firms seeking innovative change. For instance, solutions to strategically attracting and maintaining a portfolio of sustainable complements within the utilization of open innovation continue to remain mostly unanswered. This paper introduces important insights to literature pertaining to the innovation management of complements via platform policy development, specifically examining the highly influential and familiar refund policy. The focal point seeks to demonstrate the varying effects that Steam's refund policy innovation generates in games with differing attributes related to sustainability, and hence, providing preliminary understanding of decisions that platforms can make to more effectively manage their portfolio.

Incorporating the idea of cluster analysis from Bauckhage et al.'s [20] paper, games were classified into patterns of sustainability, setting a novel approach to distinguishing and prioritizing complements. The effects of the refund policy were consequently examined within each classification pattern. As aligned with Hypothesis 1, significance was shown among all patterns between a game's age and proportion of playtime distribution below 2 h, indicating significant reductions in existing revenues due to the considerable leniency of the refund policy. However, notable results were obtained from the investigation of Hypothesis 2. While the period immediately after the policy change displayed favorable outcomes, the increment of player interest quickly faded among games possessing sustainability, suggesting minimal continuation of benefits today. Instead, only unsustainable games saw sustained enhancements.

These findings suggest that Steam's remodeled policy places sustainable games in a disadvantageous position, inducing disincentives for linkage and potentially provoking reduced platform complement quality. From a macro-level of analysis, the refund policy deceptively benefits all parties, contributing to enhanced consumer rights, increased player bases, and heightened revenue for both complementors and Steam. However, from a detailed analysis, Steam's competitive advantage, primarily their proprietorship over games that generate protracted player interest, faces erosion because of ineffective platform policy management. This exemplifies the potential for mismanagement to occur, where innovations that supposedly amplify open innovation and platform leadership could become the underlying reason for its deterioration. Specifically, this study illustrates how an excessively lenient refund policy can adversely transform its portfolio of complements.

### 4.2. Theoretical Implications

Prior research has depicted limited methods concerning how the tangible innovations of platform providers can contribute to the successful development of platform dominance. The few exceptions in the literature encourage the pursuit of genre diversification [17,22] and the development of in-house complements [19]. This paper introduces several novelties that lead to vital implications for innovation management and open innovation research.

First, complements manifest distinctive characteristics which play varying degrees of contribution to platform leadership. This study exemplifies how the inventive design of Steam's refund policy can supposedly benefit all parties yet erode the prevailing platform quality. Hence, a precedent for future research is set, where the examination of platform policies and innovations should be conducted with greater consideration on complements of highest relevance. Negligence of these factors can lead to theoretical implications that appear strategically beneficial, but in truth, harvest innovative mismanagement with contrary outcomes and reductions in a platform's core advantages.

Second, this paper transitions to an operational aspect of platform policy management, exhibiting how the leniency of a refund policy can impair open innovation by hindering the development and discouraging the association of sustainable complements. Considering the prevalence of the refund policy in platform contexts, this study emphasizes the insufficiency of prior research pertaining to the refund policy, specifically in application to network settings, and proposes the necessity to incorporate additional examination at a level that integrates sustainability characteristics. This allows for a superior

understanding over the optimal construction of platform policies and consequently contributes to the emerging perception of factors around progressive innovative change.

Third, this study illustrates the significant value in focusing on platform policy within the scope of open innovation research. Platform-based markets, such as the setting that Steam operates in, presents one prevalent employment of open innovation based on the constant relationship between platform providers and complementors [2]. To succeed in such markets, the continual development and improvement of superior goods by complementors must be maintained to foster a sustainable environment. However, our findings indicate that the policies that platform providers establish or otherwise remodel could place sustainable games in a disadvantageous position. Hence, this study suggests that future open innovation research should proceed toward shedding further light in understanding and designing platform policy.

### 4.3. Managerial Implications

Previous literature has offered various managerial implications that firms can incorporate to enhance their market leadership, encompassing decisions that capitalize on the advantage of a large user and complement base [1], high platform quality [10–12], and a diversified portfolio of genres [17,22]. This paper expands these findings to platform policy management, highlighting the imperative need to understand the effects of innovative change on complements segregated by their contribution to platform leadership. More specifically, refund leniency should be designed with priority placed on ensuring that sustainable complements are provided with a robust foundation for revenue generation.

A concentration on the overarching effects of policy management, where the attributes of complements are neglected, can lead to innovative mismanagement, evoking contrary outcomes and a disintegration of competitive advantage. For instance, while platform providers may determine the benefits of platform innovations based on convincing projections and analytics, evaluations may overlook the platform's scope and source of competitive edge. Thus, instead of fixing their attention on leveraging that advantage, strategies should be devised around broadly increasing revenue. While short-term revenues may be attained, the continuous transitions to ineffective policies could provoke the selection of other platforms by disadvantaged complementors, polluting the portfolio brought about by open innovation and inducing enhanced network effects and portfolio quality for competitors. Taking this into account, caution should be applied in constructing platform policy, with emphasis placed on ensuring the preservation of the platform's future viability.

More specifically, this study illustrates the necessity in thoroughly understanding the influence of platform policy on complements at both a macro and micro level, thereby promoting superior open innovation between platform providers and complementors. For instance, this paper exemplifies how Steam's refund policy focused on fueling the macro benefits, including the enhancement of consumer rights and increase in player usage. However, this neglected the micro-level perspective, primarily the damage toward Steam's partnership over games generating extended consumer interest. This displays how unfavorable platform policy can hinder the effectiveness of open innovation, leaving a poor long-term outlook toward attracting a perpetual flow of high-value games. Consequently, the evaluation of platform policy should be conducted with a lens encompassing both a macro and micro level outlook to prevent poor strategic decisions.

### 4.4. Limitations and Future Research

This study possesses several limitations, where future research can enhance the reliability and scope of implications achieved. First, while sustainability patterns are classified in accordance with concurrent daily user patterns, little is known concerning the underlying causes for sustainable games. In other words, this paper offers insights pertaining to the attraction and preservation of a robust portfolio of complements, though increased perception in respect to the identification of characteristics among sustainable games beyond the aftermath of its player base is likewise meaningful. Future

research could examine various factors that contribute to increased complement sustainability, including software quality, marketing strategies, and gameplay aspects, among other possible determinants.

Second, special events and discounts are predominantly ignored, besides free weekends. Such events can cause varying degrees of inflation in players and owners at any point in time. Future research can analyze the properties of special events and its corresponding effects on each game's player base, allowing for the estimation and removal of the volatility induced by these events. Third, this paper classifies sustainability based on a relative assessment of each game's capability to maintain a consistent or expanding player base. However, a game's value to preserving and advancing platform leadership may involve a complex interplay of additional variables. As previous literature has highlighted, factors include the status, quality, and exclusivity of complements [17,38]. Future research can propose a comprehensive model that incorporates additional complement attributes to more appropriately distinguish complements and more effectively study strategic open innovation tactics.

Fourth, the constructed model for assessing sustainability reclassifies games that do not exceed a threshold of 0.35 in R squared to the unsustainable game pattern. Such games lack strong correspondence to the designed model, due to the extensive volatility existing within the data. Future research can devise a more robust model to classify highly volatile games. Lastly, this study exclusively analyzes the refund policy of Steam in the video game context. Hence, only general conclusions can be achieved around the design of the model and the results relating to the refund policy. Future research could employ a similar approach on platforms in various other settings to further solidify the developments achieved.

**Supplementary Materials:** The following are available online at http://www.mdpi.com/2199-8531/5/4/75/s1.

**Author Contributions:** Writing—Original Draft, S.S.; Methodology, Investigation, Software, S.S. and Y.I.; Validation, Review and Editing, S.S., Y.I. and M.T.; Resources, S.S. and M.T., Supervision, M.T.

**Funding:** This research received no external funding.

**Conflicts of Interest:** The authors declare no conflict of interest.

## Appendix A. Sufficiency of Sample Size

As previously alluded to, concerns may be raised around the small sample size among sustainable and somewhat sustainable games. The following section rationalizes the small sample size, highlighting the negligible value added from expanding the sample size. First, considering that approximately half of Steam's revenue is generated by the top 100 games [25], an analysis that incorporates the historically top 678 games encompasses most of the revenue produced. Hence, as these games are primarily responsible for Steam's profitability and platform leadership, they also serve as the primary field of interest. Furthermore, games that possess a low playtime median manifest high volatility, as depicted in Model III of Table 3, potentially invoking noise toward the identification of significant relationships. Thus, as the sample is extended toward increasingly less popular games, the volatility and consequently the standard error is likely to increase, potentially concealing correlations that could have originally been observed.

Finally, high attention is specifically paid to the strategic management of sustainable games. Figures A1 and A2 utilizes the sample employed in the analysis of refund quantities (Section 2.6) and additional revenues (Section 2.7) respectively, and illustrates the percentage of games existing in each classification pattern as each segment of the top 50 games is taken, ordered according to the quantity of owners. A trendline of best fit is placed on each constructed graph. To ensure that the presence of free weekends does not inappropriately inflate ownership data in certain games, the analysis is conducted on both 11 November and 25 November 2018. Results are similar among both tests, conveying a quickly declining logarithmic curve for the sustainable game pattern and a negatively sloped linear trendline for somewhat sustainable games. Both regressions highlight the swift decline in the number of sustainable and somewhat sustainable games discovered as increasingly less popular games are selected. Hence, the extension of the sample will likely yield minimal additional quantities

of sustainable and somewhat sustainable games, providing limited expansion to the sample size and little value to the conclusions achieved.

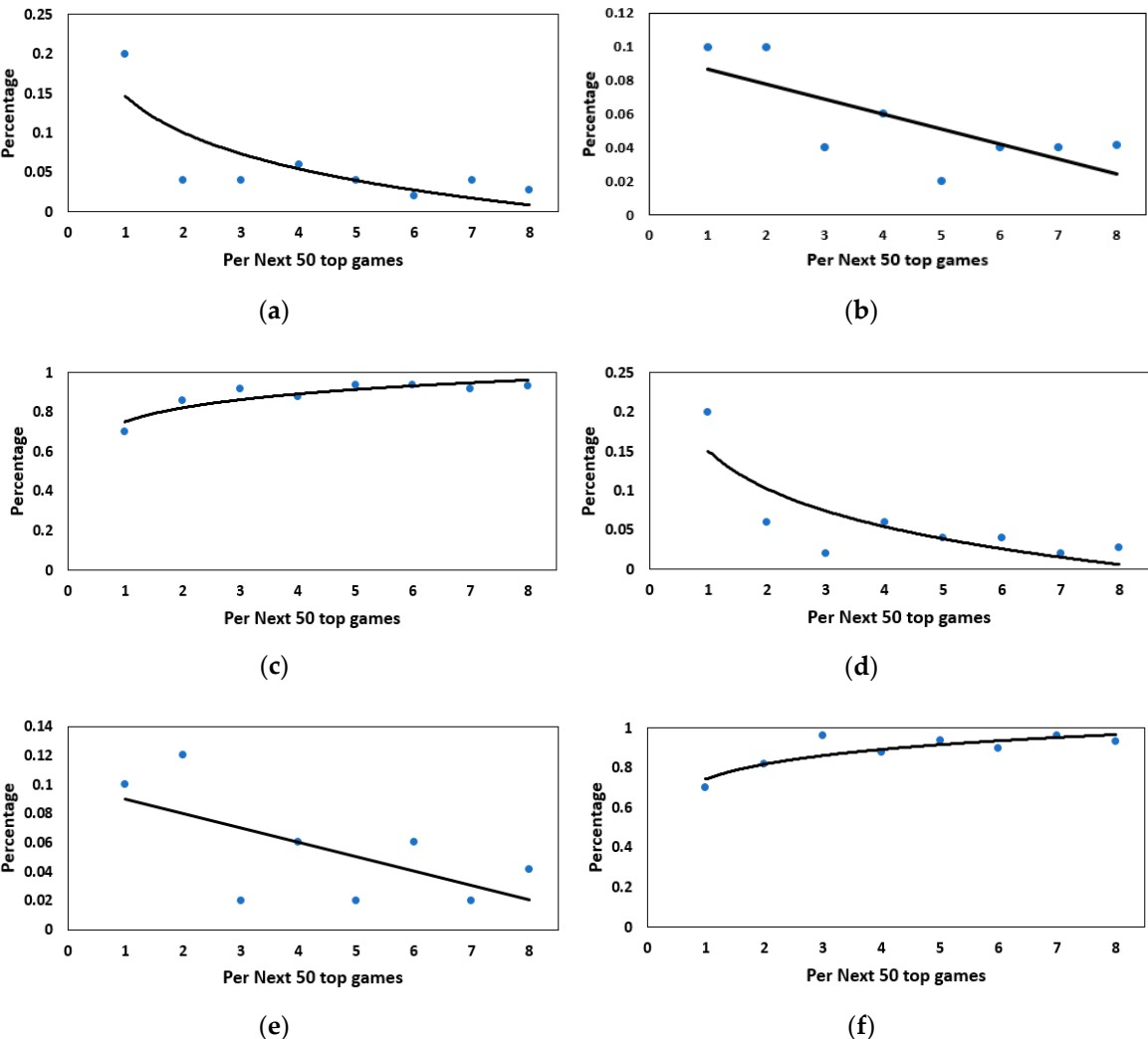

**Figure A1.** Best-fit trendline displaying relationship between percentage of each sustainability pattern per progression of 50-game popularity segment, using sample size from Section 2.6. Significance of Reductions in Existing Revenue: (**a**) Sustainable pattern—November 11 (**b**) Somewhat sustainable pattern—November 11 (**c**) Unsustainable pattern—November 11 (**d**) Sustainable pattern—November 25 (**e**) Somewhat sustainable pattern—November 25 (**f**) Unsustainable pattern—November 25.

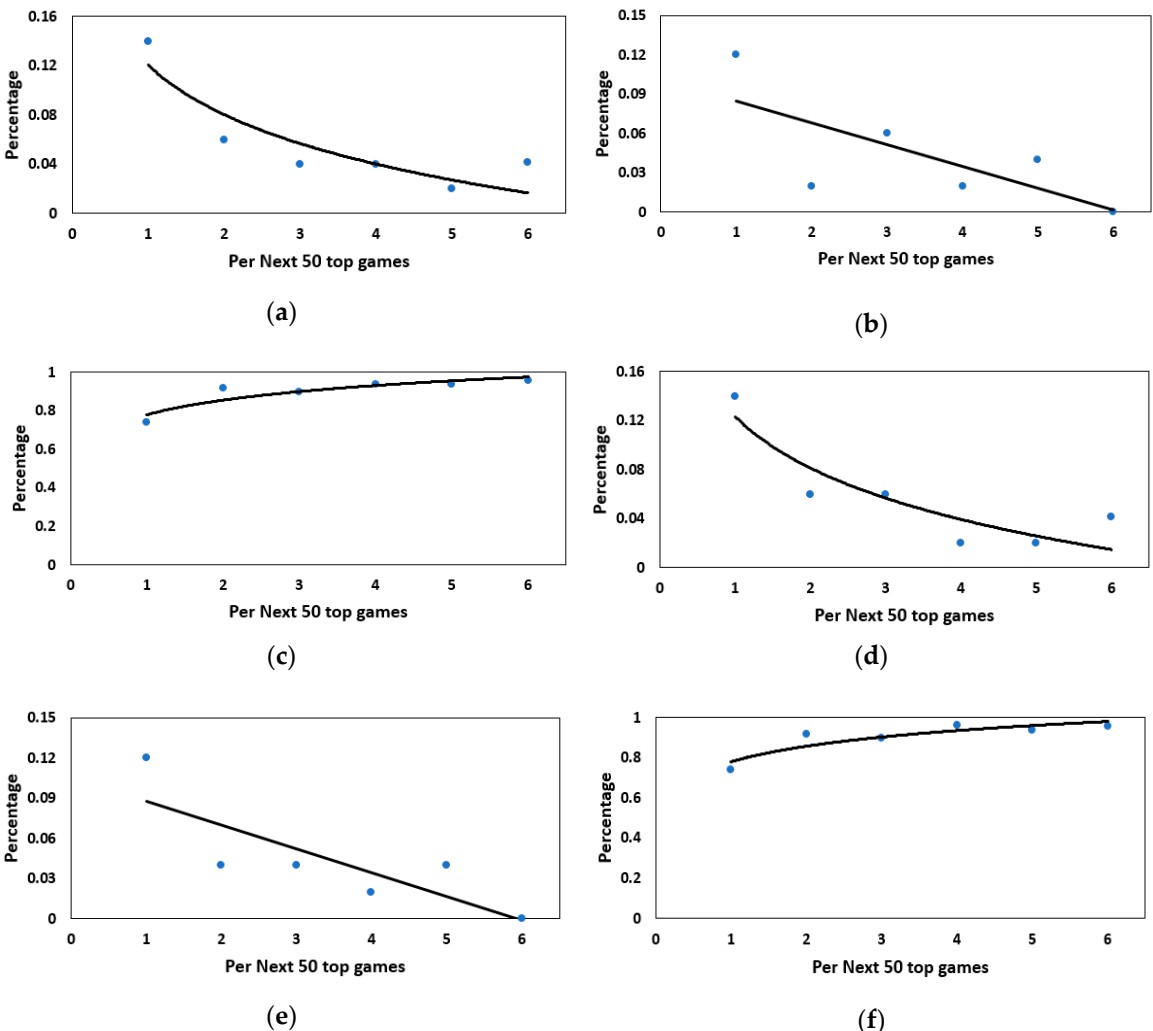

**Figure A2.** Best-fit trendline displaying relationship between percentage of each sustainability pattern per progression of 50-game popularity segment, using sample size from Section 2.7. Significance of Additional Revenue Flow: (**a**) Sustainable pattern—November 11 (**b**) Somewhat sustainable pattern—November 11 (**c**) Unsustainable pattern—November 11 (**d**) Sustainable pattern—November 25 (**e**) Somewhat sustainable pattern—November 25 (**f**) Unsustainable pattern—November 25.

## Appendix B. Additional Reclassification Procedures

*Appendix B.1. Unintended Scenarios Requiring Reclassification Procedures*

Several scenarios can lead to unintended classifications under the classification model described in Section 2.5. Classification of Sustainability Patterns. Hence, additional procedures are imposed to the preliminarily designed classification procedures to address these circumstances. For instance, three scenarios are illustrated under Figure A3, with explanations in Table A1 describing the underlying cause for each case.

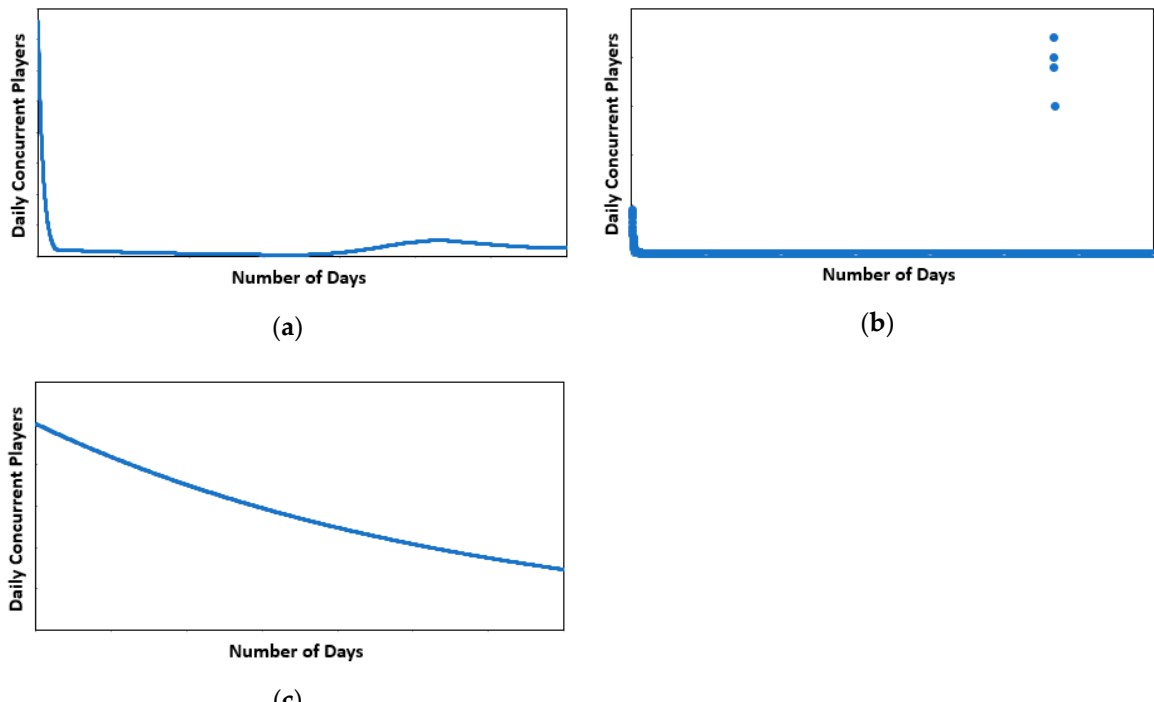

**Figure A3.** Illustrations of unintended classification scenarios requiring reclassification procedures.

**Table A1.** Explanations of Possible Unintended Classification Scenarios.

| Figure | Unintended Classification | Intended Classification | Reason |
|---|---|---|---|
| Figure A3a | Sustainable | Unsustainable | While resembling an unsustainable pattern, the slightly elevated latter half results in the latter half's average becoming higher than the former half |
| Figure A3b | Sustainable | Unsustainable | The presence of free weekends results in extreme player quantities during several days, causing the latter half's average to become higher than the former half |
| Figure A3c | Unsustainable | Somewhat Sustainable | While showing a coefficient of a > 0, the decrease is slow and steady, unlike a typical unsustainable game |

To address these issues, a set of reclassification procedures is enforced. As portrayed in Figure A3a,b, the data predominantly stays at very low measures. Hence, to identify such scenarios, the average of the top percent of all player data is calculated and a threshold is developed in accordance with half of this computation. The percentage of player data that exceeds this threshold is subsequently determined. Unlike prior classification procedures in Section 2.5, only zeros are removed from the raw data, as the elimination of low-valued data would inappropriately skew the test and heighten the percentage computed. The expected results are as follows. For unsustainable games, a low percentage should be attained, as only the initial data should exceed the threshold. Conversely, a substantial percentage should be observed in sustainable games. Table A2 proposes reclassification rules in conformance with this.

The first two reclassification procedures reallocate initially classified sustainable and somewhat sustainable games, solving issues evident in Figure A3a,b. As both issues are characterized by a low percentage surpassing the threshold, a low threshold of 5% is employed. The third reclassification rule seeks to reclassify unsustainable games where concerns illustrated in Figure A3c are present. Games that resemble a slow and steady decline tend to have a significantly higher percentage above the threshold, which is vastly contrary to the instantaneous decline of unsustainable games. This warrants reclassification to the somewhat sustainable pattern.

**Table A2.** Reclassification Procedures to Mitigate Unintended Classifications.

| Affected Group | Group to be Reclassified in | Threshold |
| --- | --- | --- |
| Sustainable | Unsustainable | < 5% |
| Sustainable | Unsustainable | < 5% |
| Unsustainable | Somewhat Sustainable | > 10% |

*Appendix B.2. Removal of Effects from Influential Free Weekends*

As evident in Figure A3b above, the presence of an extremely successful free weekend can conceal the underlying shape of the data. Under current reclassification procedures, an unsustainable game classification would essentially be guaranteed, due to the minimal percentage exceeding the threshold. This would prevail even if the underlying pattern resembled a sustainable or somewhat sustainable classification. Hence, additional procedures are executed to remove the effects of highly successful free weekends. While ideally, free weekend effects should be entirely removed, the lack of a publicly available, comprehensive list hinders this possibility. Hence, to prevent inconsistency in accounting for only discovered free weekends, procedures are enacted to specifically remove the effects of the most influential free weekends.

As with prior reclassification procedures, raw data with the exclusion of zeros is employed. A factor is then computed based on the quotient between the average of the top 3 and subsequent 3 highest data points. As significant outliers from free weekends last for 3 days, the discrepancy among the first and second set of 3 highest data points should effectively identify free weekend scenarios. However, to avoid confliction with games that quickly decline immediately after release, the first 50 data points are disregarded in calculation of the highest 6 data points. Where the factor exceeds 2.5, representing days of unnaturally high player quantities, the maximum data point as well as the two days prior to and the four days following the maximum point are eliminated for all prior analysis. While this may not account for every free weekend or the entirety of the effects, it enables the underlying data to be sufficiently unhindered and the described model to be effectively applied.

*Appendix B.3. Exceptional Reclassifications*

Despite extensive efforts to appropriately manage volatile scenarios, a small segment of games continues to fulfill the related conditions for sustainable classifications yet lack correspondence with predicted patterns. Such games tend to possess little similarity with equivalently classified games and carry low correlation with the polynomial trendline composed. Accordingly, games that do not surpass a threshold in R squared of 0.35 under the polynomial trendline constructed are reclassified to the unsustainable pattern.

The only classification exception is performed for the game, Playerunknown's Battleground (PUBG). Preservation of its original classification as a somewhat sustainable pattern is perceived as inappropriate given its unique circumstances. To further elaborate, the game has prevailed as Steam's most popular game shortly after launch. At the time of data collection, the game occupied a player base that almost doubled the runner-up game at 80 percent, despite holding only a third of its player base during its prime popularity. Its ability to sustain an unprecedented player base upholds the inappropriateness of classifying it as a somewhat sustainable pattern.

Moreover, its preliminary classification is produced by a superior average in its former half of data compared to its latter half. The underlying determinant is caused by an immediate rise to extreme popularity, before a slow and steady decline. However, unlike games that reside in the somewhat sustainable pattern, the recession of players can be primarily attributed to a drastic increase of supply in the market instead of a simple decline in player interest. To articulate further, PUBG incorporates a battle royale gameplay-style and became the first to become widely successful in this genre, paralleling the setting of a pioneer adopter. Prior to its launch, a mere 8% of core PC gamers experienced the game style. The introduction of PUBG saw a resounding 15% of core PC gamers having had played

the game within half a year into release [51]. Consequently, competition quickly capitalized the market opportunity. Notably, the free-to-play game, Fortnite, was released 4 months after PUBG and matched its popularity within 6 months, resulting in significant competition [51]. This induced a transition from a virtually competition-free environment to a highly competitive atmosphere focused on retaining traction over market share. Such circumstances assist in explaining the depression in PUBG's player base and the lack of similarity to other somewhat sustainable games, thus, asserting the inappropriateness of its initial classification.

**Appendix C. Age Distribution of Sample Size**

As the devised classification model utilizes a relative measure of game quality, concerns may be raised around the robustness of classifications. More specifically, high-aged games are at increased likelihood to reach a stage of decline or obsolescence, potentially resulting in an ongoing contemporary player base at diminished levels. The proposed model employs a comparison between the average of the former and latter half of data, with an inferior sustainability pattern placed when the first half exceeds the second. As such, mature games may be more likely to be classified toward a lower sustainability pattern, due to continuously receding values in the latter half of data. If this bias is significant, the model's approach in employing a relative measure to segregate games could be detrimentally flawed.

To ensure that this potential limitation has minimal impact and can be disregarded, a histogram analysis is conducted over the sample each classification group and compared to the distribution of the entire sample. If the bias is significant, a right-skewed and left-skewed distribution will be observed for the sustainable and unsustainable classification respectively. Conversely, if all patterns display a similarly shaped distribution, the limitation can be assumed to be negligible.

Figure A4a illustrates a comprehensive histogram containing all samples analyzed. The *x*-axis portrays the intervals of game ages, or more specifically, the number of days between the policy modification and the first day of reported data. The *y*-axis depicts the quantity of samples that are encompassed by each range. As illustrated, the construct is normally distributed around the interval bracket of 500 to 1000 days. Two games carry launch dates above 3500 days before the policy change, suggesting outlier characteristics. This is consistent with Section 3.1. High-Level Analysis of Reductions in Existing Revenue, where both data points were previously eliminated.

Figure A4b portrays the distribution of sample age for sustainable games. The histogram substantially imitates the prior comprehensive distribution (Figure A4a), including a peak around the 500- to 1000-day interval as well as minimal quantities around the first 2 brackets. Minor discrepancies can be observed, though this can be attributed to the low sample size.

Figure A4c illustrates the distribution for the sample of the somewhat sustainable pattern. The histogram paints a somewhat contrary shape, with high quantity of games in both recent and mature games, resulting in the necessity for further examination. Recall from designed classification procedures (Section 2.5) that a polynomial trendline is placed on the time-series data of concurrent daily users. Four classification scenarios encompassing the 3 sustainability patterns are possible, which are determined by 2 decision criteria, consisting of the direction of the parabola (the coefficient, a) and a comparison between the magnitudes of the former and latter half of data values. Summary procedures are provided in Table 1. The somewhat sustainable pattern is derived by only one of the four possible scenarios.

This emerges when a game adheres to an inverted parabolic trendline (a < 0), complemented by the former half of data exceeding the latter. Conversely, when the parabola is inverted but the latter half exceeds the former half, the game is considered sustainable. As previously highlighted, mature games potentially possess a tendency to gravitate toward the subordinate classification. Taking this into account, the large quantity of samples among the high-aged intervals could be elicited by this phenomenon. Precisely, mature games that hold an inverted parabolic trendline (a < 0) could be significantly biased toward the somewhat sustainable pattern, which acts as the inferior classification.

However, two reasons refute this possibility, suggesting minimal concerns in reliability for the proposed classification model. First, the distribution for sustainable games (Figure A4b) shows little evidence of skewness, suggesting that mature games containing an inverted parabolic trendline (a < 0) are not reverting to the somewhat sustainable classification. If the gravitation of mature games was significantly present, a left-skewed distribution would be expected.

Additionally, further examination can be executed on the attributes of the somewhat sustainable samples that embody the areas of concern. For instance, 8 games exist in the 2 intervals encompassing the range of 1,500 and 2,500 days. However, of these 8 games, only 2 possess an inverted parabolic trendline (a < 0), with the remaining holding an upright parabolic trendline (a > 0). This signifies that the high quantity prevalent within these 2 mature intervals cannot be caused by the tendency for older games to drift toward the inferior sustainability pattern. Rather, the contrary is proven.

To elaborate further, recall from Unintended Scenarios Requiring Reclassification Procedures (Appendix B.1) that a game is reclassified to the somewhat sustainable pattern if greater than 10% of its data values exceed a threshold equal to half of the averaged top percentage. For an obsolete game, the likelihood of surpassing this threshold would continue to diminish, due to ongoing data points that do not transcend this threshold. Despite this, the 6 games carry data characteristics that fulfill this condition, despite their protracted lifespans. In other words, an extraordinarily disproportionate quantity of mature games achieved circumstances for reclassification to a superior sustainability pattern. In view of this, the tendency for mature games to revert to an unsustainable pattern can generally be disregarded.

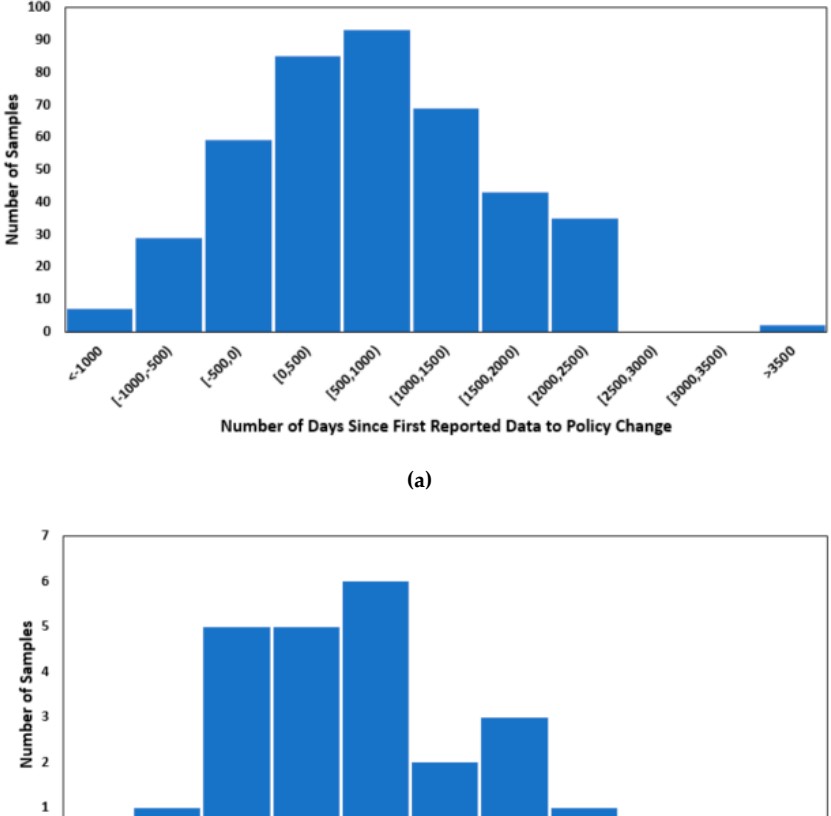

(a)

(b)

**Figure A4.** *Cont.*

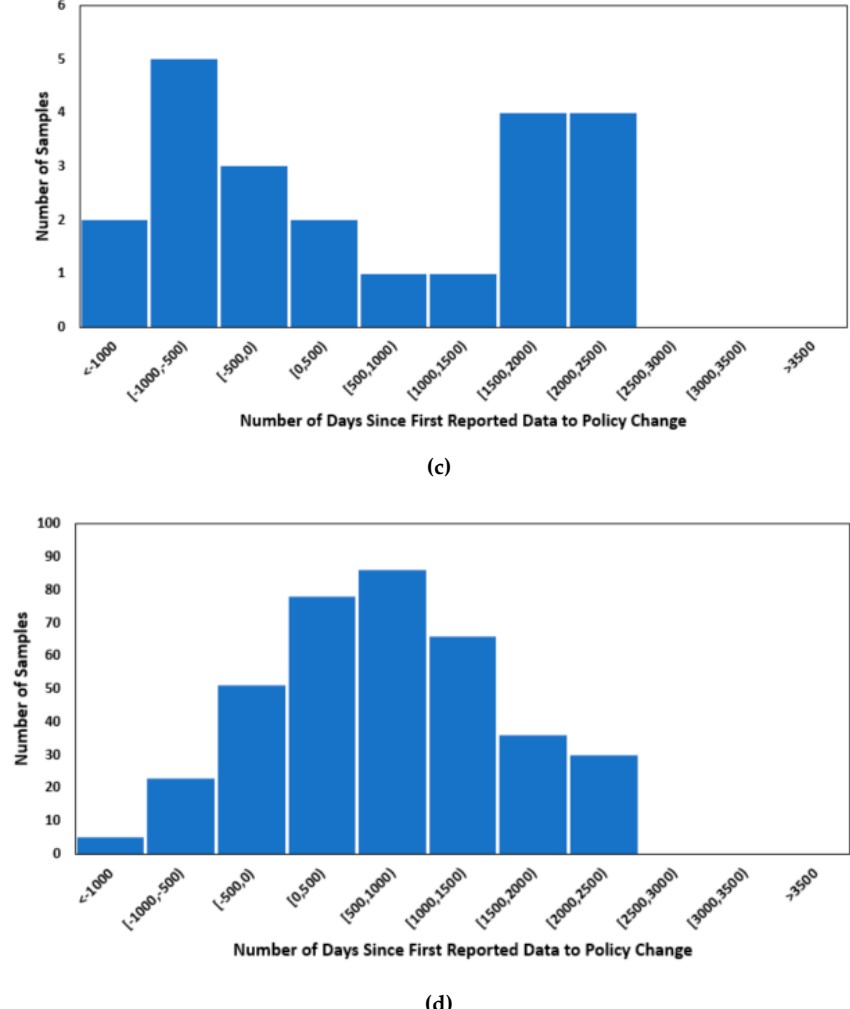

**(c)**

**(d)**

**Figure A4.** Distribution of game age for the sample utilized on each sustainability pattern, with comparison to the comprehensive age distribution containing all games analyzed: (**a**) All games (**b**) Sustainable pattern (**c**) Somewhat sustainable pattern (**d**) Unsustainable pattern.

The unsustainable pattern distribution, portrayed in Figure A4d, virtually parallels the comprehensive distribution (Figure A4a). Accordingly, this further emphasizes that there is a disregardable tendency for mature games to be inclined toward a lower sustainability pattern. The classification model is consequently considered effective and reliable under the samples utilized.

## Appendix D. Significance of Additional Revenue Flows on Individual Games

As explained in Section 2.7. Measure of Significance on Additional Revenue Flow Procedures, concerns may be raised regarding whether the impact of the policy alteration on each individual game is significantly different. Current procedures which apply the central limit theorem on binomially distributed outcomes provoke questions around whether a "favorable" outcome, where actual values exceed extrapolated values, corresponds to a negligible or significant enhancement to player and ownership quantities. For instance, the "favorable" categorization would be equivalently placed on games that are a mere 1 unit higher and games that are an entire 20 percent higher than expectations.

In addressing this, a *t*-test approach that parallels the methodology employed from Section 2.7 is executed. The *t*-test investigates the existence of significant enhancements by comparing the prior 30-day period to each 30-day period following the policy transition. The linear slope of the 30-day period prior to the policy transition is controlled for and the effects of free weekends are accommodated for in the same manner.

Hence, each favorable outcome is assessed on whether it is significantly heightened to the 5% level according to the *t*-test conducted. Table A3 summarizes the proportion of favorable outcomes that achieve significant differences to the 5% level, for each sustainability pattern as well as at a comprehensive level for both player and ownership data. Calculated percentages are exceedingly similar among all sustainability patterns, with margins of differences at around 1%. Pertaining to player data, approximately 88% of games that possess a favorable outcome show significant augmentations to the 5% significance level. Equivalently, for ownership data, 96% of favorable outcomes attain significant benefits to the 5% significance level.

These results emphasize that while procedures employ a simple binomial distribution that segregates games according to a "favorable" and "unfavorable" outcome, an extensive majority of favorable outcomes contain significantly elevated values, in response to the policy transition. This indicates that conclusions yielded by the central limit theorem depict evidence of significant magnitudes of benefits. Hence, this provides a further robustness check on the procedures encompassing the significance of additional revenues generated.

**Table A3.** Percentage of Favorable Outcomes Showing Significant Increases to 5% Level, for Each Sustainability Pattern.

|  | Sustainable | Somewhat Sustainable | Unsustainable | Comprehensive |
|---|---|---|---|---|
| Player Base Data | 88.00% | 87.50% | 88.61% | 88.55% |
| Number of Observations—Player Base Data (N) | 25 | 16 | 588 | 629 |
| Ownership Data | 95.74% | 96.55% | 96.17% | 96.16% |
| Number of Observations—Ownership Data (N) | 47 | 29 | 679 | 755 |

Note: Observation counts and calculations of percentages are comprehensive of the quantity of favorable outcomes for all three 30-day periods.

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
