# Peer review of "Erosion of Complement Portfolio Sustainability: Uncovering Adverse Repercussions in Steam’s Refund Policy"

_2199-8531, doi:10.3390/joitmc5040075_

Round 1
Reviewer 1 Report
It is an interesting and useful article. I liked reading it.
Reviewer 2 Report
The subject is interesting and relevant as the authors indicate. The article is well structured.
There are several strengths of research and paper:
The article is well structured and easy to read. The subject is contextualized appropriately and rigorously. The article proposes a clear and concise objective. The bibliography used is sufficient, relevant and updated, contextualizing the topic under study in an appropriate manner. The effort of the researchers in obtaining the interviews is valued.I present some suggestions briefly, can are useful for the authors of the article.
- In the abstract the objective must be stated very clearly. In the current state, it generates confusion. It would also be necessary to mention the work methodology clearly, as well as the results obtained.
- Better figure 1, in the current state you can not read and therefore understand.
Reviewer 3 Report
The authors have explored in this complex manuscript a case of Steam company and they combine qualitative and quantitative approaches to show how the refund policy may have adverse effects. The overall manuscript seems to be very robust and full of all necessary details and findings. Therefore, I consider it as suitable for the Journal of Open Innovation. As a reader who has rather general information about both the company and its policy (and also client experience as a gamer), I would welcome more general guidance through the main text, and more general information concerning the company (background, field, history, etc.). Therefore, I would like to encourage the authors to revise their study properly. I hope my review will be useful for the future development of this exciting paper.
First of all, the paper starts straightway from the review of the literature. I do not say that this is not possible, however, I find it quite surprising. If the authors would not mind, I would suggest adding a standard overview of the structure of the article, and information concerning the added value of the manuscript (rather in a general way).
I think that the authors should attempt to frame their paper as a case study, or as a quantitative empirical study using regression analysis and other statistical methods. However, the methods used should be introduced clearly from the very beginning.
The authors try to guide readers throughout the text, but in my opinion, further simplification and guidance are needed. It can be very simple, for example, we collect data from these sources, then use these methods to assess refund policy, and then, we conclude. The readers should very quickly understand what you are doing and how. I understand that this is very difficult, but the more complex the manuscript is the more guiding paragraphs should remind readers in the simplest way, where they are.
The conceptual overview (Figure 1) is sometimes difficult to read, the authors should be very careful when using graphics + it is not clear whether this scheme comes from their own work, or whether it was developed by other scholars (in that case, a proper citation is needed to be added).
When the authors comment on the general statistical procedures and methods, they should refer to proper literature. This is important also, when they use specific formulas, in these cases, proper page numbers are missing.
All statistical outcomes need to include a number of observations (N)
Regression models presented in Tables 3-4 lack important information concerning the description of independent variables, model statistical testing (e. g. F-test) and from Tables 5-6 it is not clear what kind of methods were used to present the data. The authors should make all Tables self-explanatory.
Finally, I would welcome more information concerning the actual research done in the abstract. The authors should explain in a very simple way, what they scientifically did, how and why.
Round 2
Reviewer 3 Report
Dear Authors,
thank you very much for the revisions you made. You did a great job while addressing my comments and I would like to congratulate on a very nice article. The only thing, which I still struggle with is the quality of the figures (i.e. Figure 1). It is still quite difficult to read the description of the boxes, and I think that the authors should provide the picture in better quality, otherwise it will be lost for the readers and that would be a big pitty.
I wish you good luck with your future research!
Your Reviewer
